# Optimal length and temporal resolution of dynamic contrast-enhanced MR imaging for the differentiation between prostate cancer and normal peripheral zone tissue

Marius Hellstern[1]*, Carlos Martinez[2], Christopher Wallenhorst[2], Dirk Beyersdorff[3], Lutz Lüdemann[4], Marc-Oliver Grimm[5], Ulf Teichgräber[6], Tobias Franiel[6]

1 Bürgerhospital und Clementin Kinderhospital gGmbH, Frankfurt am Main, Germany, 2 Institute for Epidemiology, Statistics and Informatics GmbH, Frankfurt am Main, Germany, 3 Department of Diagnostic and Interventional Radiology, University Hospital Hamburg Eppendorf, Hamburg, Germany, 4 Department of Medical Physics, Essen University Hospital, Essen, Germany, 5 Klinik und Poliklinik für Urologie Universitätsklinikum Jena, Jena, Germany, 6 Institut für Diagnostische und Interventionelle Radiologie, Universitätsklinikum Jena, Jena, Germany

* m.hellstern@buergerhospital-ffm.de

**Data Availability Statement:** All relevant data are within the paper and its Supporting Information files.

## Abstract

The value of dynamic contrast-enhanced magnetic resonance imaging (DCE-MRI) in the detection of prostate cancer is controversial. There are currently insufficient peer reviewed published data or expert consensus to support routine adoption of DCE-MRI for clinical use. Thus, the objective of this study was to explore the optimal temporal resolution and measurement length for DCE-MRI to differentiate cancerous from normal prostate tissue of the peripheral zone of the prostate by non-parametric MRI analysis and to compare with a quantitative MRI analysis. Predictors of interest were onset time, relative signal intensity (RSI), wash-in slope, peak enhancement, wash-out and wash-out slope determined from non-parametric characterisation of DCE-MRI intensity-time profiles. The discriminatory power was estimated from C-statistics based on cross validation. We analyzed 54 patients with 97 prostate tissue specimens (47 prostate cancer, 50 normal prostate tissue) of the peripheral zone, mean age 63.8 years, mean prostate-specific antigen 18.9 ng/mL and mean of 10.5 days between MRI and total prostatectomy. When comparing prostate cancer tissue with normal prostate tissue, median RSI was 422% vs 330%, and wash-in slope 0.870 vs 0.539. The peak enhancement of 67 vs 42 was higher with prostate cancer tissue, while wash-out (-30% vs -23%) and wash-out slope (-0.037 vs -0.029) were lower, and the onset time (32 seconds) was comparable. The optimal C-statistics was 0.743 for temporal resolution of 8.0 seconds and measurement length of 2.5 minutes compared with 0.656 derived from a quantitative MRI analysis. This study provides evidence that the use of a non-parametric approach instead of a more established parametric approach resulted in greater precision to differentiate cancerous from normal prostate tissue of the peripheral zone of the prostate.

**Funding:** The author(s) received no specific funding for this work.

**Competing interests:** Marius Hellstern has nothing to disclose. Christopher Wallenhorst and Carlos Martinez are employees of the Institute for Epidemiology, Statistics and Informatics GmbH. The Institute for Epidemiology, Statistics and Informatics GmbH has received grants from Bayer, Bristol-Myers Squibb and CSL Behring outside the submitted work. Marc-Oliver Grimm and Ulf Teichgräber have nothing to disclose. Tobias Franiel receives financial support from Zentrales Innovationsprogramm Mittelstand des Bundesministeriums für Wirtschaft und Energie (ZF4816001BA9), personal fees from Bayer AG, Medac GmbH and Saegeling Medizintechnik GmbH and royalties from Georg Thieme Verlag. Tobias Franiel serves on the advisory board of Bayer AG and is a member of the committee of German S3 guideline for prostate cancer. Dirk Beyersdorff has nothing to disclose. Lutz Lüdemann has nothing to disclose. This does not alter our adherence to PLOS ONE policies on sharing data and materials.

## Introduction

Prostate cancer is worldwide the second most frequent cancer in men accounting for 14.5% of new cancers in men, 1.4 million new cases and 375,000 deaths in 2020. The age-standardized incidence rate varies across regions from 6.3 per 100,000 in south-Central Asia to 83.4 per 100,000 in Northern Europe [1].

Multiparametric magnetic resonance imaging (MRI), comprises T1- and T2-weighted imaging, diffusion-weighted imaging (DWI) and dynamic contrast-enhanced (DCE) MRI, has been established as a non-invasive imaging modality for detecting and staging of localized prostate cancer lesions [2–5]. However, the value of DCE MRI in the detection of prostate cancer is still controversial. A meta-analysis has shown the usefulness of parametric DCE-MRI in the differential diagnosis of prostate cancer and non-cancerous tissue in the peripheral zone and central gland [6]. Parametric DCE-MRI is complex and time intensive as based on several assumptions. There are currently insufficient peer reviewed data or expert consensus to support routine adoption of DCE-MRI for clinical use [7].

Non-parametric DCE-MRI analysis is less complicated and time-consuming than parametric DCE-MRI as acquisition requirements are less extensive (e.g. no need for the so-called arterial input function measurement) and measurement values estimation is performed directly. In contrast to parametric approaches, non-parametric approaches are more suited for fast and simple non-invasive image-based diagnostics. However, the optimal measurement conditions are not yet standardized across centres.

The aim of this study was to evaluate the optimal temporal resolution and measurement length for DCE-MRI to differentiate cancerous from normal prostate tissue of the peripheral zone of the prostate by non-parametric MRI analysis and to compare with quantitative MRI analysis.

## Methods

### Study design, setting and subjects

This study was a secondary cross-sectional analysis of data collected from an original work aimed at differentiating healthy tissue from prostate cancer in the peripheral zone using DCE-MRI parameters, published elsewhere [8].

The study cohort consisted of patients with biopsy-proven prostate cancer who underwent prostate MRI before radical prostatectomy between November 2004 and October 2008. Patients with presence of a cardiac pacemaker or other electronic implant, reported claustrophobia, known allergy to gadolinium-based contrast agents, lymph node metastasis, cardiovascular risks, known renal insufficiency or failure to give written informed consent were excluded.

**MRI protocol.** Patients underwent DCE-MRI (1.5-T whole-body MRI system, Magnetom Sonata, Siemens Medical Systems, Erlangen, Germany) preoperatively with 1.5-T with a body coil for radio frequency excitation, two elements of a Spine-Array-Coil, two elements of a Body-Phased-Array-Coil and a endo rectal coil (Medra, Pittsburgh, PA, USA) were combined for data acquisition.

Conditions of measurement were defined according to Prochnow et al 2005 [9]. The evaluation was performed by using 21 transversal T2-weighted images and 513 phase-images of each patient, S1 Fig.

**Handling of radical prostatectomy specimens.** Details of the histopathologic analysis and its correlation with cancer tissue and normal prostate tissue have been published previously [10, 11]. In summary, an experienced pathologist prepared and assessed all prostatectomy specimens. Slices were perpendicular to the long axis of the gland for optimal correlation

with the T2-weighted images and the DCE-MRI slice, which were also axially angulated perpendicular to the long axis and the DCE-MRI slice. Slice orientations and corresponding paraffin blocks were recorded on a chart. Together with a radiologist, the pathologist selected paraffin blocks that matched the DCE-MRI slices. For this they used a pathologic diagram, coronal T2-weighted images and sagittal localizer images. The urethra was used as a landmark for aligning axial T2-weighted images with their corresponding paraffin blocks. The selected blocks were cut into 4 μm sections and stained with haematoxylin and eosin.

**Postprocessing DCE-MRI datasets.** The signal intensity time curves (S(t)) were calculated by using a dedicated prostate MRI biopsy software DynaCAD® version: 2.1.7.113583 for the processing of data. The region of interest (ROI) relevant position on T2-weighted image was marked with a cursor. The position was synchronized with the phase image prior plotting the signal intensity time curves. Depending on the size of the ROI 2 measuring points in 57 specimens and 3 measuring points in 40 specimens within the ROI were determined. These data were used for calculating the signal intensity time curve. The results were presented in a coordinate system indicating the time in seconds on the horizontal axis and the intensity values on the vertical axis. The function of mean evaluation was used for noise suppression for each individual measurement. The intensity of the signal was obtained each 1.6 seconds for a total period of 820.8 seconds.

These data were the basis for further temporal resolution and measuring length investigations. Furthermore, quantitative parameters were derived from parametric maps [12]. For further data collection, the data visualization software Amira® version: 5.3.3 was used. The relevant ROI on the T2-weighted image was drawn and synchronized with parametric maps for mean blood volume, mean transit time, mean perfusion, and mean extravascular volume.

## Terms and measures

Various settings were investigated based on the measuring length and the temporal resolution of the DCE-MRI. There were 16 measuring lengths of interest: 2, 2.5, 3, 3.5, 4, 4.5, 5, 5.5, 6, 6.5, 7, 7.5, 8, 10, 12 and 13 minutes. The seven temporal resolutions of interest were $1.6^{-1}$, $3.2^{-1}$, $4.8^{-1}$, $6.4^{-1}$, $8.0^{-1}$, $16.0^{-1}$ and $30.4^{-1}$ measurements per second. All 112 combinations of measured length and temporal resolution (16 measurement lengths and 7 temporal resolutions) were investigated. The outcome of interest was prostate cancer area in the peripheral zone compared with normal prostate tissue of the peripheral prostate zone. Prostatitis tissues were not analysed.

The covariates of interest were derived from intensity-time-profile of the DCE-MRI. The intensity-time-profile was described using a non-parametric analysis, by obtaining the following summary measurement variables as representatives for the intensity-time-profile: (1) time between injection of contrast agent and start of enhancement phase [seconds] ($T_0$) (onset time), (2) ratio of intensity at maximum and at start of enhancement [%] ($\frac{Sm}{S0}$) (relative signal intensity, RSI), (3) average increase in signal intensity during enhancement phase [intensity/second] ($\frac{Sm-S0}{Tp-T0}$) (wash-in slope), (4) maximum intensity ($S_m$) (peak enhancement), (5) intensity decrease between time of intensity maximum and end of recorded period [%] ($\frac{Sm-Sfinal}{Sm}$) (wash-out), and (6) average decrease in signal intensity during reduction phase [intensity/second] ($\frac{Sm-Sfinal}{Tmax-Tp}$) (wash-out slope).

We used the first minute after the onset time for the determination of the wash-in parameters (RSI, wash-in slope and peak enhancement) [13].

RSI: relative signal intensity; $S_{final}$: last point of the signal-intensity curve; $S_m$: maximum signal intensity; $S_0$: intensity at start of enhancement; $T_0$: onset time; $T_{max}$: measurement length; $T_p$: time to reach maximum signal intensity.

For the comparison with the quantitative method, the following quantitative parameters were investigated: mean blood volume, mean transit time, mean perfusion, and mean extravascular volume.

## Data analysis

For each of the 112 settings the following analyses were conducted and a score derived to predict prostate cancer: First, a multivariate logistic regression was performed using prostate cancer as a dichotomous dependent variable and the 6 covariates as continuous independent variables. A stepwise selection of independent variables was then performed using Wald tests and removing independent variables with $p > 0.2$. A score to predict prostate cancer was then defined as the sum of logits of the remaining independent variables in the final model.

Second, to estimate the discriminatory power of this score, i.e. the ability to separate subjects who have a prostate cancer area from those who have not, the following approach was performed: a C-statistics based on cross validation using the leave one pair out method were calculated [14]. The cross validation was performed as follows: (a) selection of a subject with prostate cancer area (case) and a subject without prostate cancer area (control) from the group of all subjects, (b) application of the score development methodology on the remaining subjects, as described above, (c) calculation of the score for the selected case and the selected control, (d) repetition of steps (a-c) for all possible combinations of cases and controls, and (e) calculation of the C-statistics as the relative frequency of concordant pairs, using the number of all possible pairs as denominator. Pairs were assumed to be concordant, if the score for the case was greater than the score for the control.

Third, to test the robustness of the score, a sensitivity analysis without restricting the wash-in period to one minute was conducted as was an analysis using the same independent variables in the logistic regression models for all settings without subsequent stepwise reduction, based on the set of independent variables in the final model of the setting with the highest C-statistics. All groups of C-statistics were analysed for a linear trend depending on measurement length and temporal resolution using linear regression models.

For comparison, we calculated mean values of blood volume, transit time, perfusion, and extravascular volume in normal and prostate cancer tissues respectively. We performed a multivariate logistic regression with stepwise selection of independent variables using prostate cancer as a dichotomous dependent variable and defined a score to predict prostate cancer as the sum of logits of the remaining independent variables in the final model, and calculated a C-statistics based on cross validation using the leave one pair out method.

All analyses were performed using Stata MP version 14.2 (StataCorp LLC).

## Research ethics standards compliance

The secondary research data collected for this study at the Charité, Berlin were completely and robustly anonymised. Based on Thuringian law and our clinical Ethics Committee, reports providing routinely collected data do not need approval by the Ethics Committee.

## Patient and public involvement

There was no public or patient involvement in the conception of the research question or the design or implementation of the study.

## Results

Of 61 patients with biopsy-proven prostate cancer, seven patients were excluded for the following reasons: two with non-available histology, two with non-available signal intensity time

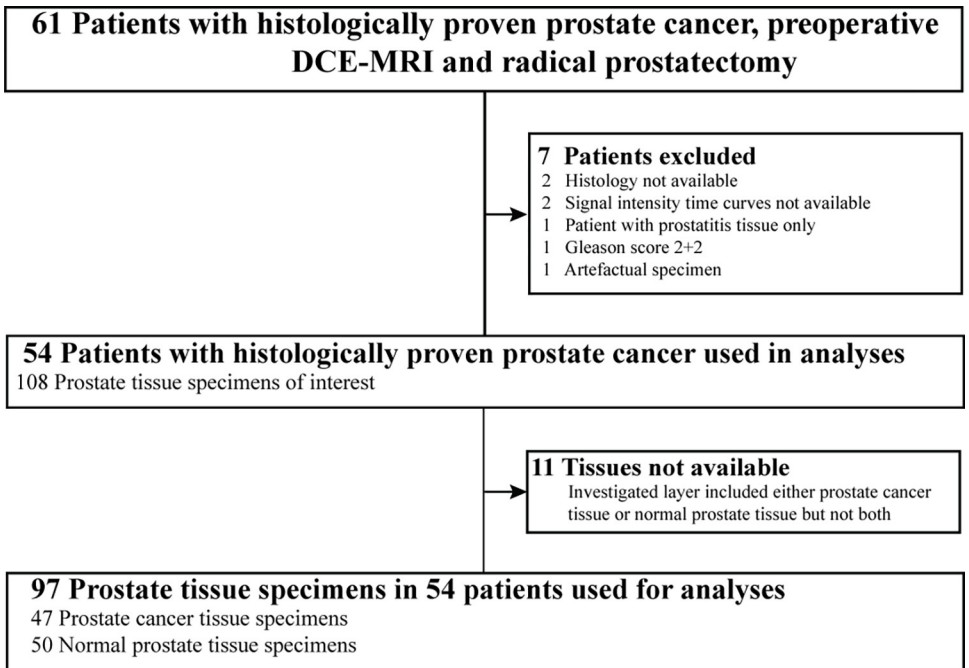

**Fig 1. Flowchart for inclusion and exclusion for patient and tissue selection.** DCE-MRI: dynamic contrast-enhanced MRI.

curves, one with prostatitis tissue only, i.e. neither normal nor cancer tissue, one with inappropriate pathological Gleason-score 2+2 and one with artefactual specimen. A total of 97 tissue specimens were provided from radical prostatectomy, 47 assessed as prostate cancer and 50 as normal prostate tissue (43 patients with cancer and normal tissue, 4 with cancer tissue only and 7 with normal prostate tissue only), Fig 1. Analysed were 54 patients with biopsy-proven prostate cancer who underwent prostate MRIs (mean age at MRI: 63.8 years, range: 49-71years), mean time between MRI and total prostatectomy 10.5 days (range: 1–49 days). After surgery the tumour had a Gleason score 6 in 15 patients and 39 patients had Gleason score ≥7, Table 1.

A total of 10,864 signal intensity time curves were generated by using 16 measurement lengths and 7 temporal resolutions for each specimen.

The distribution of the 6 summary measurement variables of the signal intensity time curves stratified by tissue type is presented in Fig 2. When comparing prostate cancer tissue with normal prostate tissue, the median RSI (422% vs 330%), wash-in slope (0.870 vs 0.539) and peak enhancement (67 vs 42) were greater respectively, while the wash-out (-30% vs -23%) and wash-out slope (-0.037 vs -0.029) were smaller, and the onset time (32 seconds) similar.

C-statistics from cross validation of the ability to separate prostate cancer area from normal prostate area varied between 0.669 and 0.743 when all combinations of the 7 resolutions and 16 measurement lengths for the DCE-MRI were investigated, Table 2.

Similar ranges were seen in the sensitivity analyses, S1 and S2 Tables in S1 File.

The best model for the differentiation of normal prostate from prostate cancer tissue in the base analysis with respect to C-statistics was achieved with a temporal resolution of 8 seconds and measurement length of 2.5 minutes. The corresponding C-statistics was 0.743 and the model comprised RSI, wash-in slope and wash-out slope as prediction covariates, Table 3.

In the sensitivity analysis without restriction of the wash-in period to 1 minute, the best C-statistics (0.750) related to a temporal resolution of 8.0 seconds, a measurement length of 13 minutes and wash-in slope as the only covariate, Table 3.

**Table 1. Patient demographics and histopathological data.**

| Clinical/histopathological data | Data |
|---|---|
| **Total** | 54 |
| **Age at MR imaging [Years]** | |
| Mean (SD) | 63.8 (5.3) |
| Median (p25,p75) | 65 (61,68) |
| Min, Max | 49, 71 |
| **Time between MRI and total prostatectomy [days)]** | |
| Mean (SD) | 10.5 (9.7) |
| Median (p25,p75) | 8 (4,13) |
| Min, Max | 1, 49 |
| **PSA at time of diagnosis [ng/ml]** | |
| Mean (SD) | 18.9 (80.7) |
| Median (p25,p75) | 7 (5,9) |
| Min, Max | 1, 600 |
| **Gleason-Score at prostatectomy, n (%)** | |
| 3+3 | 15 (27.8) |
| 3+4 | 20 (37.0) |
| 4+3 | 5 (9.3) |
| 4+4 | 9 (16.7) |
| 4+5 | 5 (9.3) |

PSA: prostate specific antigen; MRI: magnetic resonance imaging.

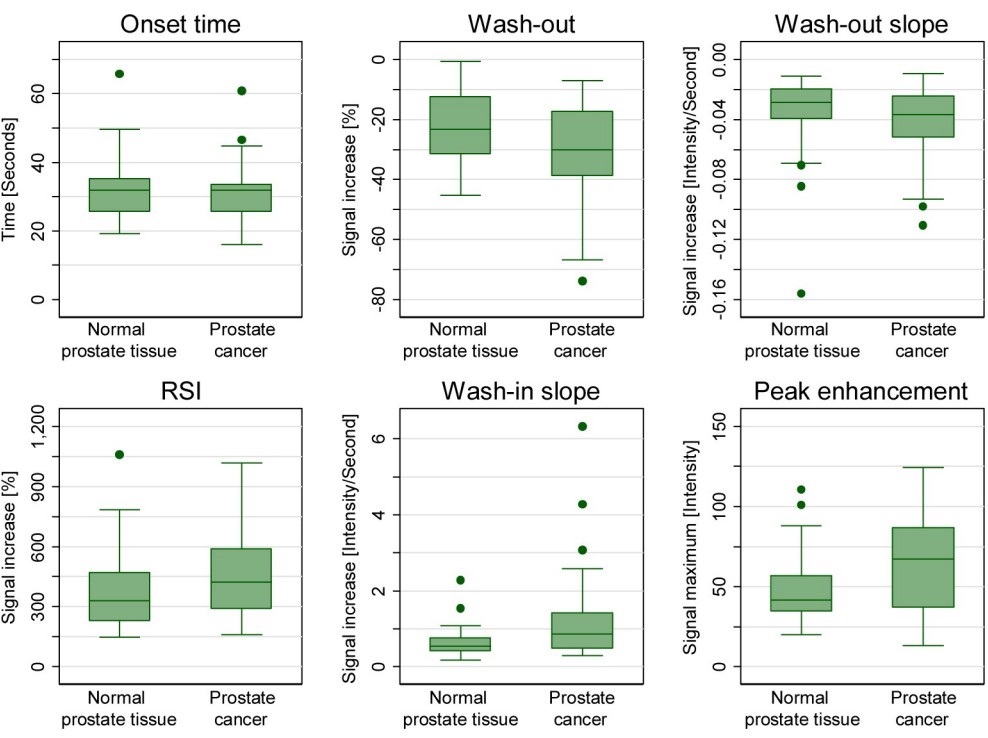

**Fig 2. Distribution of parameters from the raw data.** RSI: relative signal intensity.

**Table 2. C-statistics from cross validation by measurement length and temporal resolution–base analysis.**

| Measurement length [Minutes] | Temporal resolution [Seconds] | | | | | | |
|---|---|---|---|---|---|---|---|
| | 1.6 | 3.2 | 4.8 | 6.4 | 8.0 | 16.0 | 30.4 |
| 2.0 | 0.708 | 0.694 | 0.680 | 0.676 | 0.715 | 0.711 | 0.713 |
| 2.5 | 0.712 | 0.700 | 0.719 | 0.699 | 0.743 | 0.734 | 0.713 |
| 3.0 | 0.709 | 0.695 | 0.680 | 0.699 | 0.716 | 0.712 | 0.713 |
| 3.5 | 0.709 | 0.700 | 0.680 | 0.722 | 0.716 | 0.712 | 0.713 |
| 4.0 | 0.709 | 0.719 | 0.681 | 0.720 | 0.725 | 0.712 | 0.694 |
| 4.5 | 0.709 | 0.707 | 0.717 | 0.702 | 0.740 | 0.712 | 0.713 |
| 5.0 | 0.709 | 0.691 | 0.680 | 0.735 | 0.724 | 0.708 | 0.703 |
| 5.5 | 0.709 | 0.681 | 0.680 | 0.718 | 0.716 | 0.712 | 0.671 |
| 6.0 | 0.709 | 0.694 | 0.680 | 0.699 | 0.716 | 0.712 | 0.716 |
| 6.5 | 0.709 | 0.696 | 0.698 | 0.718 | 0.716 | 0.712 | 0.713 |
| 7.0 | 0.741 | 0.683 | 0.680 | 0.699 | 0.716 | 0.712 | 0.704 |
| 7.5 | 0.717 | 0.702 | 0.690 | 0.714 | 0.716 | 0.720 | 0.713 |
| 8.0 | 0.701 | 0.731 | 0.697 | 0.699 | 0.716 | 0.714 | 0.714 |
| 10.0 | 0.709 | 0.695 | 0.685 | 0.697 | 0.731 | 0.723 | 0.695 |
| 12.0 | 0.709 | 0.693 | 0.680 | 0.704 | 0.722 | 0.715 | 0.713 |
| 13.0 | 0.724 | 0.685 | 0.669 | 0.708 | 0.716 | 0.712 | 0.713 |

The impact of measurement length and temporal resolution on C-statistics was analysed using linear trend analysis. The base model showed a non-significant increase in C-statistics depending on additional second of temporal resolution. The sensitivity analysis based on the best model yielded a significant C-statistics increase of 0.000445 (95%-confidence interval: 0.000004 to 0.000886) for every additional second of temporal resolution. Increase of measurement length did not provide a significant change of C-statistics in the base and all sensitivity analyses, S3 Table in S1 File.

In normal prostate tissue, mean blood volume (0.007 versus 0.010%), mean extravascular volume (0.240% versus 0.307%) and mean perfusion (0.357 versus 0.702 mL/min/cm$^3$) were lower than in prostate cancer tissue, whereas mean transit time was higher in normal tissue than in prostate cancer tissue (2.819 versus 2.303 seconds). After applying the stepwise approach, the score to predict prostate cancer based on quantitative parameters included the

**Table 3. Best models according to C-statistics from cross validation.**

| Measurement length [Minutes] | Temporal resolution [Seconds] | Covariates used in final model | Cross validation [C-statistic] |
|---|---|---|---|
| Base analysis: With restricting Wash-In period | | | |
| 2.5 | 8.0 | RSI, wash-in slope, wash-out slope | 0.743 |
| 7.0 | 1.6 | Wash-in slope, wash-out slope | 0.741 |
| 4.5 | 8.0 | RSI, wash-in slope, wash-out | 0.740 |
| 5.0 | 6.4 | Peak enhancement, wash-out | 0.735 |
| 2.5 | 16.0 | Wash-in slope, wash-out slope | 0.734 |
| Sensitivity analysis: Without restricting Wash-In period | | | |
| 13.0 | 8.0 | Wash-in slope | 0.750 |
| 5.0 | 6.4 | Peak enhancement, wash-out | 0.735 |
| 8.0 | 3.2 | Peak enhancement, wash-out, wash-out slope | 0.731 |
| 4.5 | 8.0 | Peak enhancement, wash-out | 0.731 |
| 4.0 | 8.0 | Peak enhancement, wash-out | 0.730 |

variables blood volume (increasing risk with increasing volume) and transit time (increasing risk with decreasing time). The C-statistics of this model was 0.656 compared to 0.743 of the non-parametric semiquantitative model.

## Discussion

The discriminatory power to distinguish between prostate cancer and normal prostate tissue using the non-parametric DCE-MRI analysis varied widely with temporal resolution, measurement length and included covariate combination. Considering the benefit of short measurement length, the optimal C-statistics of 0.743 was achieved for a temporal resolution of 8 seconds and measurement length of 2.5 minutes using the variables RSI, wash-in slope and wash-out slope. In contrast, the discriminatory power of the quantitative method yielded a C-statistics of 0.656. Without restricting the wash-in period to the first minute, the C-statistics was slightly larger (+0.007) but required an increase in measurement length from 2.5 to 13 minutes. RSI, wash-in slope and wash-out slope predicted the presence of prostate cancer. In contrast, increasing measurement length did not lead to higher discrimination power due to relevant phase of wash-in is done in first minute [15–17].

### Interpretation in context of the literature

To the best of our knowledge, there is no study relating the optimal temporal resolution and measurement length in DCE-MRI. However, the Prostate Imaging—Reporting and Data System version 2.1 (PIRADSv2.1) recommends a temporal resolution of <15 seconds and a total observation rate ≥2 min [7]. We provide evidence for best discrimination using temporal resolution of 8 seconds and measurement length of 2.5 minutes, which is in line with the recommendations of the American college of Radiology, the European Society of Uroradiology (ESUR), and the AdMetech Foundation [7]. We identified RSI, wash-in slope, wash-out slope as combination of covariates with highest discrimination power, these results are in line with previous studies, that showed, that wash-in parameters were useful for tumour detection in the transitional zone [16, 18]. Furthermore, Wash-in slope has been a very useful parameter for prostate cancer detection and localization [18–21] probably due to increased blood flow and neoangiogenesis in the tumour [22]. However a temporal resolution of >10 seconds is resulting in a significant reduction of the diagnostic accuracy of wash-in parameters [23].

There is evidence that simple descriptive parameters are sufficient to distinguish prostate cancer from normal prostate tissue. In some cases, there is evidence that descriptive parameters are superior to pharmacokinetic parameters [16, 24]. The ROC curves of pharmacokinetic parameters were stated as being "only fair discriminators" (k(ep): 0.6; K(trans): 0.6 and v(e): 0.56).[10] Pharmacokinetic parameters have the capability to distinguish between tumour and benign peripheral zone prostatic tissue, but simple descriptive parameters have a greater sensitivity and specificity [25].

### Clinical/Pathophysiologic implications

Biparametric MRI (T2-weighted imaging and DWI) has been proposed as a good alternative to achieve optimal accuracy to detect prostate cancer without DCE-MRI, especially in patients with impaired renal function. Two recent meta-analyses of head-to head comparisons of the accuracy for detecting prostate cancer with multiparametric MRI and biparametric MRI of 11 T2-weighted imaging and 20 DWI studies respectively showed similar specificity of detecting prostate cancer [26, 27]. However, the sensitivity was significantly higher for multiparametric MRI on one of the two only [26]. Advantages of biparametric MRI include shorter scan time periods in the machine resulting in increased patient comfort and compliance, no use of MRI

contrast agents and therefore no contrast agent-associated side effects such as nephrogenic systemic fibrosis, and reduced costs [28]. Large scale studies may be needed to prove that biparametric MRI is not inferior to multiparametric MRI [29].

DCE-MRI is used in clinical practise in particular when DWI is diminished by artefact or if there is an inappropriate signal-to-noise-ratio in DWI [7]. Concerning staging procedures DCE-MRI may give a more precise representation of the volume of the index prostate cancer in comparison to T2-weighted imaging or DWI-sequences [30]. DCE-MRI remains substantial for patients with suspected local recrudescence after prior treatment, like radical prostatectomy or radiotherapy [31, 32].

The parametric and non-parametric methods for DCE-MRI analysis have good correlations and are widely accepted approaches [13, 24, 25, 33]. The non-parametric approach is an appropriate simplified method for DCE-MRI analysis with the advantage of less required assumptions. Our results of specifying the optimal temporal setting will allow a more standardized non-parametric approach which support in clinical practice a clear distinction between patients with prostate cancer and patients with normal prostate tissue. Furthermore, there is the possibility to minimize the appearance of artefacts due to patient movement, rectal peristaltic and continuous bladder filling with reduced measurement length.

## Strengths and limitations

The main strength of this study is the greater precision for evaluating the optimal temporal resolution and measurement length for DCE-MRI to differentiate cancerous from normal prostate tissue of the peripheral zone of the prostate. This was achieved by a non-parametric approach which is based on fewer assumptions than parametric approaches.

We assume that healthy tissue of patients with prostate cancer is comparable to healthy tissues of patients without prostate cancer regarding the estimated effect of the temporal resolution and measuring length on the specificity of DCE-MRI. Furthermore, this is a retrospective study prone to bias due to unmeasured confounding. The limited number of patients and consequently of the study power could have resulted in non-identification of additional relevant prediction parameters, and consequently in reduced C-statistics.

## Conclusions

In conclusion, evaluation of DCE-MRI intensity-time-profiles based on non-parametric summary parameters is a useful method for the principal clinical needs of prostate cancer localisation and staging. Our study has shown that DCE-MRI settings of 2.5 minutes measurement length and 8.0 seconds temporal resolution are most appropriate to identify prostate cancer in the peripheral zone. In future studies, the validity of our recommended measurement approach should be investigated in external patient cohorts preferentially along with a comparison to parametric approaches in the same patient cohort.

## Supporting information

**S1 File. File includes S1-S3 Tables.**
(DOCX)

**S1 Fig. DCE-MRI: Dynamic contrast-enhanced MRI.**
(TIF)

**S1 Dataset.**
(TXT)

**S2 Dataset.**
(TXT)

**S3 Dataset.**
(TXT)

## Acknowledgments

I would like to thank DB and LL for the data collection, which made this work possible.

## Author Contributions

**Conceptualization:** Marius Hellstern, Marc-Oliver Grimm, Ulf Teichgräber, Tobias Franiel.

**Data curation:** Dirk Beyersdorff, Lutz Lüdemann.

**Formal analysis:** Marius Hellstern, Carlos Martinez, Christopher Wallenhorst.

**Methodology:** Marius Hellstern, Carlos Martinez, Christopher Wallenhorst.

**Supervision:** Tobias Franiel.

**Visualization:** Carlos Martinez, Christopher Wallenhorst.

**Writing – original draft:** Marius Hellstern.

**Writing – review & editing:** Marius Hellstern, Carlos Martinez, Christopher Wallenhorst, Dirk Beyersdorff, Lutz Lüdemann, Marc-Oliver Grimm, Ulf Teichgräber, Tobias Franiel.

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
