## [Decision Letter · Decision Letter 0]

10 Jan 2023

PONE-D-22-32433Optimal length and temporal resolution of dynamic contrast-enhanced MR imaging for the differentiation between prostate cancer and normal peripheral zone tissuePLOS ONE

Dear Dr. Hellstern,

Thank you for submitting your manuscript to PLOS ONE. After careful consideration, we feel that it has merit but does not fully meet PLOS ONE’s publication criteria as it currently stands. Therefore, we invite you to submit a revised version of the manuscript that addresses the points raised during the review process.Provide the details of the radiological pathological correlation and how they differentiated prostate cancer tissue and normal prostate tissue.Please provide other prostate MRI findings in addition to the DCE-MRI.Please compare the non-parametric semiquantitative method with the quantitative method.Please submit your revised manuscript by Feb 24 2023 11:59PM. If you will need more time than this to complete your revisions, please reply to this message or contact the journal office at plosone@plos.org. Please include the following items when submitting your revised manuscript:A rebuttal letter that responds to each point raised by the academic editor and reviewer(s). You should upload this letter as a separate file labeled 'Response to Reviewers'.A marked-up copy of your manuscript that highlights changes made to the original version. You should upload this as a separate file labeled 'Revised Manuscript with Track Changes'.An unmarked version of your revised paper without tracked changes. You should upload this as a separate file labeled 'Manuscript'.

We look forward to receiving your revised manuscript.

Kind regards,

Quan Jiang, Ph,D.

Academic Editor

PLOS ONE

Journal Requirements:

"I have read the journal's policy and the authors of this manuscript have the following competing interests:

Marius Hellstern has nothing to disclose. Christopher Wallenhorst and Carlos Martinez employees of the Institute for Epidemiology, Statistics and Informatics GmbH. The Institute for Epidemiology, Statistics and Informatics GmbH has received grants from Bayer, Bristol-Myers Squibb and CSL Behring outside the submitted work. Marc-Oliver Grimm and Ulf Teichgräber have nothing to disclose. Tobias Franiel receives financial support from Zentrales Innovationsprogramm Mittelstand des Bundesministeriums für Wirtschaft und Energie (ZF4816001BA9), personal fees from Bayer AG, Medac GmbH and Saegeling Medizintechnik GmbH and royalties from Georg Thieme Verlag. Tobias Franiel serves on the advisory board of Bayer AG and is a member of the committee of German S3 guideline for prostate cancer. Dirk Beyersdorff has nothing to disclose. Lutz Lüdemann has nothing to disclose."

Reviewers' comments:

Reviewer's Responses to Questions

**Comments to the Author**

1. Is the manuscript technically sound, and do the data support the conclusions?

Reviewer #1: Partly

Reviewer #2: Partly

2. Has the statistical analysis been performed appropriately and rigorously? 

Reviewer #1: Yes

Reviewer #2: Yes

3. Have the authors made all data underlying the findings in their manuscript fully available?

Reviewer #1: Yes

Reviewer #2: Yes

4. Is the manuscript presented in an intelligible fashion and written in standard English?

Reviewer #1: Yes

Reviewer #2: Yes

5. Review Comments to the Author

Reviewer #1: Thank you very much for selecting me as a reviewer for the article “Optimal length and temporal resolution of dynamic contrast-enhanced MR imaging for the differentiation between prostate cancer and normal peripheral zone tissue”.

This paper deals with very interesting topic of dynamic contrast-enhanced magnetic resonance which is an insufficiently used method and there is a great scientific interest in increasing usability of this method. Problem with the DCE-MRI is lack of standardisation between institutions.

The authors stated that “There are currently insufficient peer reviewed published data or expert consensus to support routine adoption of DCE-MRI for clinical use“ which is only partially correct. DCE-MRI is included in all PI RADS version. According to currently valid PI RADS v2.1 DCE-MRI is used as quantitative method so it is in clinical use. There is insufficient data on semiquantitative and quantitative DCE-MRI, that is , there is a need for research that presents the superiority of these two methods and research that deals with the standardisation of these methods.

Also the authors, through the abstract, introduction and discussion, conclude about the superiority of the non-parametric semiquantitative method compared to the quantitative method, however, as they stated in the limitations of the research, they did not compare the two methods.

There are many research, not only regarding prostate but also many other organ systems, published in high quality journals that support use of quantitative method od DCE-MRI wherever possible. Quantitative study by observing the movement of contrast medium between intra- and extravascular space gives information about tumour angiogenesis so it has a role not only in locoregional staging of the disease or potential role in PI RADS categorisation but also in monitoring of oncologic treatment response.

Methodology is explained in detail, which is the major strength of this manuscript in order to guide further research on this topic. The research is conceptually designed and developed with high quality.

The results are certainly of great value in directing further research.

I hope my comment would be of help for the authors to make their study more suitable for publication.

Thank you.

Reviewer #2: The aim of the study is assessing the optimal temporal resolution and measurement lenght for

DCE-MRI for differentiating peripheral prostate cancer and normal prostate gland. The

authors emerged that the study cohort is consisted of patients with who underwent prostate

MRI before radical prostatectomy. They use onset time, relative signal intensity (RSI), wash-

in slope, peak enhancement, wash-out and wash-out slope as predictors of interest that

determined from non-parametric characterisation of DCE-MRI intensity- time profiles. A C-

statistics based on cross validation were performed to esmimate the discriminatory power.

The Authors stated that DCE-MRI settings of 2.5 minutes measurement length and 8.0

seconds temporal resolution are most appropriate to identify prostate cancer in the peripheral

zone.

The strong aspects of this study is as follows:

- The aim of this study is to evaluate for optimal DCE-MRI parameters, which is a

simple and fundamental need.

- There is no study in the literature evaluating optimal remporal resolution and

measurement length.

However, this study has some major shortcomings.

- To give optimal values, the number of study chort that was conducted from 54 patients

with 97 prostate tissue specimens (47 prostate cancer, 50 normal prostate tissue) of the

peripheral zone is insufficient.

- DCE-MRI is not a dominant sequence in prostate MRI evaluation and should always

be read in conjunction with the other MRI sequences. However, the authors do not

mention other prostate MRI findings in the article.

- The authors did not explain how they made the radiological pathological correlation

and how they differentiated prostate cancer tissue and normal prostate tissue.

6. PLOS authors have the option to publish the peer review history of their article (what does this mean?). If published, this will include your full peer review and any attached files.

Reviewer #1: No

Reviewer #2: No

---

## [Author Response · Author response to Decision Letter 0]

18 Apr 2023

Response to Reviewers

Reviewer comment #1: Provide the details of the radiological pathological correlation and how they differentiated prostate cancer tissue and normal prostate tissue.

Authors’ response: 

Thank you for bringing this to our attention. Details of the radiological pathological correlation and how they differentiated prostate cancer tissue and normal prostate tissue of the study subjects their specimens have been published previously. However, our summary of the “Handling of radical prostatectomy specimens” in the methods section has not referenced earlier publications with the details requested. 

Following the reviewer’s comment we have expanded the description of the handling of radical prostatectomy specimens in the methods section.

Added text in methods section:

“Details of the histopathologic analysis and its correlation with cancer tissue and normal prostate tissue have been published previously.(1, 2) In summary, an experienced pathologist prepared and assessed all prostatectomy specimens. Slices were perpendicular to the long axis of the gland for optimal correlation with the T2-weighted images, which were also axially angulated perpendicular to the long axis and the DCE-MRI slice. Slice orientations and corresponding paraffin blocks were recorded on a chart. Together with a radiologist, the pathologist selected paraffin blocks that matched the DCE-MRI slices. For this they used a pathologic diagram, coronal T2-weighted images and sagittal localizer images. The urethra was used as a landmark for aligning axial T2-weighted images with their corresponding paraffin blocks.” The selected blocks were cut into 4 µm sections and stained with haematoxylin and eosin, supplementary figure 1.

Reviewer comment #2: Please provide other prostate MRI findings in addition to the DCE-MRI.

Authors’ response: We appreciate this suggestion. Other findings in T2W images include areas of chronic prostatitis with band-like, wedge-like, or diffuse T2w hypointensity with corresponding inflammatory lymphocyte and plasma cell infiltrates. However, areas of chronic prostatitis were not further investigated as the aim of this study was to differentiate cancerous from normal prostate tissue.

No changes to the manuscript were applied.

Reviewer comment # 3: Please compare the non-parametric semiquantitative method with the quantitative method.

Authors’ response: Thank you for this comment. As requested by the reviewer, we have conducted a comparison of the non-parametric semiquantitative method with quantitative parameters from each of the 97 prostate tissue specimens. Quantitative parameters in prostate cancer and normal prostate tissues respectively were blood volume, mean transit time, perfusion, and extravascular volume. We applied the same statistical approach as for the non-parametric semiquantitative method to predict prostate cancer diagnosis. We performed a multivariate logistic regression model with stepwise selection of independent variables (the 4 quantitative parameters) and prostate cancer as a dichotomous dependent variable, developed and derived a score to predict prostate cancer as the sum of logits of the remaining independent variables in the final model, and calculated a C-statistics based on leave-one-pair-out cross-validation.

In normal prostate healthy tissue, mean blood volume (0.007 versus 0.010%), extravascular volume (0.240 versus 0.307%) and perfusion (0.357 versus 0.702 mL/min/cm³) were lower than in prostate cancer tissue, whereas mean transit time was higher in normal prostate tissue than in prostate cancer tissue (2.819 versus 2.303 seconds).

After applying the stepwise approach, the score to predict prostate cancer based on quantitative parameters included the variables blood volume (increasing risk with increasing volume) and transit time (increasing risk with decreasing time). The C-statistics of this model was 0.656 compared to 0.743 of the non-parametric semiquantitative model.

The comparison of the non-parametric semiquantitative method with the quantitative method required the following manuscript changes.

Abstract, study objective:

Thus, the objective of this study was to explore the optimal temporal resolution and measurement length for DCE-MRI to differentiate cancerous from normal prostate tissue of the peripheral zone of the prostate by non-parametric MRI analysis “and to compare with a quantitative MRI analysis.

Abstract, last sentence of study results:

The optimal C-statistics by non-parametric MRI analysis was 0.743 for temporal resolution of 8.0 seconds and measurement length of 2.5 minutes “compared with 0.656 derived from a quantitative MRI analysis.”

Introduction, last paragraph:

The aim of this study was to evaluate the optimal temporal resolution and measurement length for DCE-MRI to differentiate cancerous from normal prostate tissue of the peripheral zone of the prostate by non-parametric MRI analysis “and to compare with quantitative MRI analysis”.

Methods, added text at end of description of Postprocessing of DCE-MRI Datasets:

“Furthermore, quantitative parameters were derived from parametric maps.(3) For further data collection, the data visualization software Amira® version: 5.3.3 was used. The relevant ROI on the T2-weighted image was drawn and synchronized with parametric maps for mean blood volume, mean transit time, mean perfusion, and mean extravascular volume.”

Methods, subheading Terms and measures: 

“For the comparison with the quantitative method, the following quantitative parameters were investigated: mean blood volume, mean transit time, mean perfusion, and mean extravascular volume.”

Data analysis, last paragraph, added text: 

“For comparison, we calculated mean values of blood volume, transit time, perfusion, and extravascular volume in normal and prostate cancer tissues respectively. We performed a multivariate logistic regression with stepwise selection of independent variables using prostate cancer as a dichotomous dependent variable and defined a score to predict prostate cancer as the sum of logits of the remaining independent variables in the final model, and calculated a C-statistics based on cross validation using the leave one pair out method.”

Results, last paragraph, added text:

“In normal prostate tissue, mean blood volume (0.007 versus 0.010%), mean extravascular volume (0.240% versus 0.307%) and mean perfusion (0.357 versus 0.702 mL/min/cm³) were lower than in prostate cancer tissue, whereas mean transit time was higher in normal tissue than in prostate cancer tissue (2.819 versus 2.303 seconds).

After applying the stepwise approach, the score to predict prostate cancer based on quantitative parameters included the variables blood volume (increasing risk with increasing volume) and transit time (increasing risk with decreasing time). The C-statistics of this model was 0.656 compared to 0.743 of the non-parametric semiquantitative model.”

Discussion, first paragraph, added text:

The discriminatory power to distinguish between prostate cancer and normal prostate tissue using the non-parametric DCE-MRI analysis varied widely with temporal resolution, measurement length and included covariate combination. Considering the benefit of short measurement length, the optimal C-statistics of 0.743 was achieved for a temporal resolution of 8 seconds and measurement length of 2.5 minutes using the variables RSI, wash-in slope and wash-out slope. “In contrast, the discriminatory power of the quantitative method yielded a C-statistics of 0.656.” 

References

1. Franiel T, Ludemann L, Rudolph B, Lutterbeck E, Hamm B, Beyersdorff D. Differentiation of prostate cancer from normal prostate tissue: role of hotspots in pharmacokinetic MRI and histologic evaluation. AJR Am J Roentgenol. 2010;194(3):675-81.

2. Franiel T, Ludemann L, Rudolph B, Rehbein H, Stephan C, Taupitz M, et al. Prostate MR imaging: tissue characterization with pharmacokinetic volume and blood flow parameters and correlation with histologic parameters. Radiology. 2009;252(1):101-8.

3. Ludemann L, Prochnow D, Rohlfing T, Franiel T, Warmuth C, Taupitz M, et al. Simultaneous quantification of perfusion and permeability in the prostate using dynamic contrast-enhanced magnetic resonance imaging with an inversion-prepared dual-contrast sequence. Ann Biomed Eng. 2009;37(4):749-62.

---

## [Decision Letter · Decision Letter 1]

12 Jun 2023

Optimal length and temporal resolution of dynamic contrast-enhanced MR imaging for the differentiation between prostate cancer and normal peripheral zone tissue

PONE-D-22-32433R1

Dear Dr. Hellstern,

We’re pleased to inform you that your manuscript has been judged scientifically suitable for publication and will be formally accepted for publication once it meets all outstanding technical requirements.

Kind regards,

Lorenzo Faggioni, M.D., Ph.D.

Academic Editor

PLOS ONE

Additional Editor Comments (optional):

Thank you for your reply.

Reviewers' comments:

Reviewer's Responses to Questions

**Comments to the Author**

1. If the authors have adequately addressed your comments raised in a previous round of review and you feel that this manuscript is now acceptable for publication, you may indicate that here to bypass the “Comments to the Author” section, enter your conflict of interest statement in the “Confidential to Editor” section, and submit your "Accept" recommendation.

Reviewer #1: All comments have been addressed

2. Is the manuscript technically sound, and do the data support the conclusions?

Reviewer #1: Yes

3. Has the statistical analysis been performed appropriately and rigorously? 

Reviewer #1: Yes

4. Have the authors made all data underlying the findings in their manuscript fully available?

Reviewer #1: Yes

5. Is the manuscript presented in an intelligible fashion and written in standard English?

Reviewer #1: Yes

6. Review Comments to the Author

Reviewer #1: Dear Authors and Editor,

I am very satisfied with the corrections that authors have made and with detailed answers they have given to the reviewer’s comments.

Sincerely.

7. PLOS authors have the option to publish the peer review history of their article (what does this mean?). If published, this will include your full peer review and any attached files.

Reviewer #1: No

---

## [Editor Report · Acceptance letter]

16 Jun 2023

PONE-D-22-32433R1 

Optimal length and temporal resolution of dynamic contrast-enhanced MR imaging for the differentiation between prostate cancer and normal peripheral zone tissue 

Dear Dr. Hellstern:

I'm pleased to inform you that your manuscript has been deemed suitable for publication in PLOS ONE. Congratulations! Your manuscript is now with our production department. 

Kind regards, 

on behalf of

Dr. Lorenzo Faggioni 

Academic Editor

PLOS ONE